

# Regional variation in fire weather controls the reported occurrence of Scottish wildfires

G. Matt Davies[1] and Colin J. Legg[2]

[1] School of Environment and Natural Resources, The Ohio State University, Columbus, OH, United States of America
[2] School of GeoSciences, The University of Edinburgh, Edinburgh, Scotland, United Kingdom

## ABSTRACT

Fire is widely used as a traditional habitat management tool in Scotland, but wildfires pose a significant and growing threat. The financial costs of fighting wildfires are significant and severe wildfires can have substantial environmental impacts. Due to the intermittent occurrence of severe fire seasons, Scotland, and the UK as a whole, remain somewhat unprepared. Scotland currently lacks any form of Fire Danger Rating system that could inform managers and the Fire and Rescue Services (FRS) of periods when there is a risk of increased of fire activity. We aimed evaluate the potential to use outputs from the Canadian Fire Weather Index system (FWI system) to forecast periods of increased fire risk and the potential for ignitions to turn into large wildfires. We collated four and a half years of wildfire data from the Scottish FRS and examined patterns in wildfire occurrence within different regions, seasons, between urban and rural locations and according to FWI system outputs. We used a variety of techniques, including Mahalanobis distances, percentile analysis and Thiel-Sen regression, to scope the best performing FWI system codes and indices. Logistic regression showed significant differences in fire activity between regions, seasons and between urban and rural locations. The Fine Fuel Moisture Code and the Initial Spread Index did a tolerable job of modelling the probability of fire occurrence but further research on fuel moisture dynamics may provide substantial improvements. Overall our results suggest it would be prudent to ready resources and avoid managed burning when FFMC > 75 and/or ISI > 2.

Corresponding author
G. Matt Davies, davies.411@osu.edu

## INTRODUCTION

Globally, fire is one of the preeminent disturbances controlling the structure of ecosystems and the services they deliver (*Archibald et al., 2013*). Although wildfires are a natural component of many ecosystems, in others they are associated with human land-use and management of fire regimes has played a critical role in the development of many valuable ecosystems (e.g., *Webb, 1998*). Wildfires can, however, also pose significant threats to ecological, economic and cultural resources and there is growing concern over how

wildfire activity may be altered by a changing climate (e.g., *Scholze et al., 2006*; *Doerr & Santín, 2016*). Many countries invest significant resources in fire management and wildfire fighting with attention increasingly on adapting fire regimes rather than outright fire suppression (e.g., *Peterson, Halofsky & Johnson, 2011*; *Stephens et al., 2013*). A critical part of any response system for managing wildfire is the need to be able to plan for periods of significant wildfire activity and safely complete managed burning activities. Substantial research has therefore been focused on the assessment of global (e.g., *Scholze et al., 2006*), national (e.g., *Tanskanen & Venäläinen, 2008*; *de Jong et al., 2016*) and regional (e.g., *Dimitrakopoulos, Bemmerzouk & Mitsopoulos, 2011*; *Beccari et al., 2015*) assessments of climatic drivers of wildfire activity. Modeling wildfire activity remains challenging both due to the quality of historical fire data available from documentary records and satellite data (*Murphy et al., 2000*; *Krawchuk & Moritz, 2014*), and the statistical challenges involved (e.g., *Andrews, Loftsgaarden & Bradshaw, 2003*; *Eastaugh, Arpaci & Vacik, 2012*; *de Jong et al., 2016*). In addition our understanding of wildfire regimes remains incomplete in many regions even though the environmental imperative for improving knowledge is significant. In northern ecosystems, for example, there is very significant concern about potential feedbacks between climate change, wildfire activity and severity, and ecosystem carbon dynamics (*Dorrepaal et al., 2009*; *Turetsky et al., 2015*).

Fire is an integral part of the ecology of the British uplands. Traditional managed burning is used extensively for habitat management for red grouse (*Lagopus lagopus scoticus* Latham 1787) on heather (*Calluna vulgaris* (L.) Hull) dominated moorlands and blanket bogs, and to rejuvenate moorland and grassland (principally where dominated by purple moor grass, *Molinia caerulea* (L.) Moench, in the latter case) for cattle, sheep and deer grazing (*Thompson et al., 1995*). In forests prescribed fire has also been used as a ground preparation tool prior to planting (*Aldhous & Scott, 1993*) and to facilitate restoration in native woodlands (*Hancock et al., 2009*). Whilst there continues to be substantial debate about the environmental costs and benefits of managed burning, particularly in relation to the effect of fire on carbon dynamics (e.g., *Glaves et al., 2013*; *Davies et al., 2016a*), land-managers, conservationists and government agencies are increasingly aware of the potential for severe wildfires to cause substantial environmental damage (e.g., *Maltby, Legg & Proctor, 1990*; *Davies et al., 2013*). Wildfires are a common occurrence in grass and shrub dominated moorland vegetation and in gorse (*Ulex europaeus* L.) stands close to urban areas (*Legg et al., 2007*). Wildfires within forests in Scotland are much less common, though they do occur during exceptional weather conditions and in young plantations of conifers, especially where these are adjacent to heather or grass-dominated vegetation, or where heather has re-invaded older stands after thinning (*Aldhous & Scott, 1993*). Although there are reports in the UK of naturally-occurring wildfires associated with lightning ignitions (e.g., *Allison, 1954*), the British climate means that in most years a very high proportion of, if not all, wildland fires are of anthropogenic origin initiating as accidental fires, as escaped management burns, or from arson.

Wildfire activity is widely expected to increase across the British uplands and increased wildfire activity and severity is recognised as one of the more significant threats to UK biodiversity (*Sutherland et al., 2008*). These projected trends are driven by: fuel
**Table 1  Description of the fuel moisture codes and fire behavior indices of the Canadian Fire Weather Index system.**

| Code/Index | Description |
| --- | --- |
| Fine Fuel Moisture Code (FFMC) | Moisture content of cured leaves, needles and small dead twigs on the forest floor |
| Duff Moisture Code (DMC) | Moisture content of loosely-compacted, partially decomposed needle litter |
| Drought Code (DC) | Moisture content of deep layers of compact humus and organic matter |
| Build-up Index (BUI) | Weighted combination of DMC and DC designed to represent total fuel available for combustion |
| Initial Spread Index (ISI) | Combines FFMC and wind speed to provide representation of potential rate of spread |
| Fire Weather Index (FWI) | Weighted combination of ISI and BUI designed to provide representation of potential fireline intensity |

accumulation associated with changes in sheep stocking rates (*Acs et al., 2010*); pressure to reduce the extent of, or even ban, managed burning (*Backshall, Manley & Rebance, 2001*; *Davies et al., 2016a*); climate change predictions suggesting summers will become warmer and drier with more frequent droughts (*Jenkins et al., 2009*); and increased ignition frequencies associated with widening public land access (Land Reform (Scotland) Act 2003) and in England and Wales (Countryside & Rights of Way Act 2000, the so-called CROW Act). The economic cost of wildfires in the UK cannot be easily estimated but they are likely to be substantial. The costs include the destruction of property (forestry, fencing, etc.), lost income from reduced land productivity, and the costs of suppression for land-managers and local agencies (*Farmer, 2003*; *Joint Arson Group, 2007*). The environmental costs of wildfire can also be considerable, particularly where peat is ignited resulting in destruction of the seedbank, a higher risk of erosion and a complete change in ecosystem function (e.g., *Maltby, Legg & Proctor, 1990*; *Davies et al., 2013*).

A coherent approach to developing wildfire management policy is slowly emerging in the UK but a robust Fire Danger Rating System is still needed (*Gazzard, McMorrow & Aylen, 2016*). The difficulties and costs associated with development of such a system are considerable. For example, in New Zealand, fire behaviour and fuel moisture in similar shrub-dominated fuel types and temperate climates have proven to be challenging to model (*Alexander, 2008*). Globally a number of countries have investigated or adopted the Canadian Fire Weather Index System (FWI System; Table 1) to provide forecasts of wildfire danger (e.g., *DaCamara et al., 2014*; *Simpson et al., 2014*). A limited system currently exists in the form of the Met Office Fire Severity Index (MOFSI; *Kitchen et al., 2006*) developed for England and Wales in response to the CROW Act. MOFSI is based on the FWI system which recent research suggests has promise for detecting variation in fire risk across the UK despite important regional variation in temporal patterns in FWI system codes/indices (*de Jong et al., 2016*). MOFSI's single five-point index is not sufficient to capture variation in weather and fuel conditions that are of interest to those who work with vegetation fires in the UK or who are responsible for responding to wildfire events.

At present The Met Office do not make the underlying codes and indices of the FWI system available, though the full suite of outputs are available from the European Forest Fire Information System which provides up to six-day forecasts on a 10-km grid for the whole of Europe. Previous research in the UK has suggested that the Duff Moisture Code and Drought Code of the FWI system (relating respectively to the moisture content of partially decomposed and deep organic matter) show promise in forecasting fire severity (*Davies et al., 2013*; *Davies et al., 2016b*), and that the Fine Fuel Moisture Code (FFMC) may be a tolerable predictor of the wildfire activity (*Legg et al., 2007*).

This paper's aims to develop guidelines for forecasting potential wildfire activity by using wildfire occurrence data from four regions of the Scottish Fire and Rescue Service (see Supplemental Information 1) to meet three objectives: (i) compare the results of differing methodologies that can be used to assess the performance of fire risk indices; (ii) describe broad temporal and spatial patterns of wildfire occurrence in Scotland; and (iii) model the relationship between wildfire occurrence, event magnitude and fuel moisture codes and fire weather indices provided by the FWI system. We acknowledge that, as *Finney (2005)* pointed out, wildfire occurrence data do not allow us to model wildfire probability *per se*. The constraints on our data are discussed below. Nevertheless, a better understanding of the relationship between wildfire occurrence and fire weather conditions has the potential to contribute significantly to fire management where there is currently no information at all.

## MATERIALS & METHODS

### Wildfire incidents

Securing accurate information on wildfire activity has, until very recently, not been a simple task. There is no central repository for such information and different regional Fire and Rescue Services (FRS), though nominally all using the "Incident Reporting System" (*DCLG, 2012*), held their own information and recorded wildfire data that varied considerably in scope and quality. Furthermore, we know from experience that the FRS by no means attends all wildfires with call-out rates likely to vary significantly across the UK. For instance in rural, upland areas, where traditional managed burning is still prevalent, gamekeepers and land-managers have historically been reluctant to involve the FRS. In such locations reporting of fires is likely to be incomplete for social reasons and due to sheer remoteness. The data available is therefore probably biased towards smaller burns in accessible and/or more urban areas.

We received data on 4,343 wildfire incidents from four former regional Fire Authorities across Scotland (Supplemental Information 1, Fig. S1). The data covered the period between 2003 and spring 2007 though this varied somewhat by region. For the purposes of analysis the fire records were restricted to those within the period between 1st January 2003 and 15th March 2007. This provides a minimum 30-day lead-in time for calculation of the FWI system codes/indices for which Met Office data were available (see below). The data from the Lothian & Borders FRS only ran to August 2006, so the overall data set does not have equal spatial coverage of all years/months and the data for Lothian & Borders is slightly biased towards the spring and summer.

The content of the data provided was very variable ranging from simple records of date and location to more detailed descriptions that included the personnel and equipment used to fight the fire, the burn area and the vegetation type burnt. Interpretation was made more complex by the fact that the four regions used different terminology for describing the fire and resources used. Although most records came with a descriptor of fuel type, this was imprecise and inconsistent. Thus "moorland fire" and "heath fire" might include vegetation dominated by heather or grass whilst the term 'grass fire' appeared, in some cases, to be used for wildfires in general, including those in heather and gorse. The records also included fuel types referred to as crop and stubble, woodland, forest, bushes, hedges and gorse. Fires included in a category described as other (including bonfires, vehicles, rubbish or unspecified) were excluded for this analysis. We classified each of the fire locations as being urban or rural, where urban records were defined in our database as those occurring within the periphery of the 100 largest towns and cities in Scotland.

Information about the magnitude of fires varied from estimates of the area burned to precise measures of the number of man-hours involved. The magnitude of each incident was scored by assigning a score from 0–5 for duration of the incident (between report time and all F&RS returning to base), the area burnt, the number of appliances in attendance, the number of "resources" used (assumed to be the number of F&RS people in attendance), "Number in attendance" (it is not clear if this included non-F&RS personnel) and number of person hours. The scoring system is described in detail in *Legg et al. (2007)*. The different data sets presented very different types of information and it was not possible to analyse these data directly so the maximum of the above values for any one fire was recorded as the overall measure of that event's magnitude. There were rather few fires with a maximum score of 4 so the final index was created by merging classes 3 and 4. This formed an arbitrary 5-point scale where 0 indicates that the incident was small by all counts and ≥4 indicates that the incident was in about the top 10% with respect to one or more of duration, area burned or resources deployed. For subsequent analysis we defined fires with a final score of 4 or 5 as "large fires." We acknowledge that this measure is rather crude but until such time as we have available data for a number of years with consistent recording of fire area, fire severity or fire behaviour it is the best currently available. From mapping the locations of the fires it was apparent that, for a substantial number of the records, the location recorded was where the fire was reported from or where fire-fighting resources were marshalled rather than the core or ignition point of the fire itself. We assume this will not have substantial impact on the accuracy of FWI system values assigned as these are calculated at a broad spatial scale (see below). Combined with the lack of accurate information on vegetation burnt it did, however, mean that we were unable to account for variation in fuel type at this stage. Previous authors have noted similar issues regarding historical fire records in other studies (e.g., *Murphy et al., 2000*; *Krawchuk & Moritz, 2014*) but given the quantity of data available and the broad spatial scale at which analysis was completed we were confident of being able to detect important fire weather signals from our admittedly noisy data.

## Fire weather data

Indices and codes of the FWI system were calculated for the dates and locations of all recorded wildfires. Data used for the calculation of the FWI system were provided by the Met Office from their Numerical Weather Prediction now-casting model in NIMROD data format. Data represented the predicted weather variables at the centre of a 5 km grid-square on the Ordnance Survey National Grid. A detailed description of all the weather data received and analysed is provided in *Legg et al. (2007)*. We chose to use this data for three reasons: (i) data from widely-distributed observational stations was unlikely to be representative of conditions at the fire ground due to complex terrain; (ii) it represented the form and quality of data that would be used in any implemented version of a fire danger rating system; and (iii) this data is also used to drive the Met Office Fire Severity Index and its spatial resolution is favourable compared to previous, similar studies in Europe (e.g., *Padilla & Vega-García, 2011*).

The six standard components of the FWI system (FFMC, DMC, DC, BUI, ISI and FWI; Table 1) were calculated using the equations given in *Van Wagner & Pickett (1985)*. Met Office data were available from 1st December 2002 so the minimum lead-in time for fires in January 2003 was 31 days.

## Data analysis

The distribution of wildfires is non-random in both time and space. In particular, fuel hazard in spring, with abundant dead herbaceous and aerial shrub fuels, is quite different from that in summer where most above-ground fuel is live. Additionally grass fuels are more abundant in the north-west of Scotland. Fires are more frequent in spring and summer, but can occur at any time of year. To test the discriminating power of the fire weather indices we therefore required a set of weather data that represent "typical" weather conditions, independent of the occurrence of fire, but which had the same spatial and temporal distribution as the weather records for locations and days on which fires were reported. Thus, for each fire, the FWI system indices were calculated for a "control day" in the same grid square on the same calendar day but in the year prior to the fire. For fires in 2003 the calculations were for the same day in 2006. This provides a baseline distribution of values for each index against which the characteristics of fire days can be compared. We acknowledge that our control days are pseudo-absences and it is impossible to rule out the possibility that unreported fires occurred in the control dates and locations. We also acknowledge that wildfires in our region are often ignition limited as rapidly-spreading, intense fire behaviour has been shown to occur even at very low ISI and FWI values in our shrubland fuel types (*Davies et al., 2006*). We worked under the hypothesis that the FWI system would still capture useful variation in fire weather conditions that might reflect the potential for wildfires due to escaped burns and increased propensity for accidental ignitions during particularly dry conditions.

Our analysis followed a two stage process where we first screened the outputs of the FWI system to identify those codes or indices that appeared to discriminate best between fire and control days. Initially, we examined conditional probability curves for each output, calculated using the "cdplot" function in R 3.1.2 (*R Core Team, 2014*), and histograms of

the distributions of fire day and control day output values. Further scoping followed the approaches described by *Andrews, Loftsgaarden & Bradshaw (2003)* and *Eastaugh, Arpaci & Vacik (2012)*. We used the Mahalanobis distance (Eq. (1); *Viegas et al., 1999*) to examine the ability of the FWI system outputs to discriminate between fire days and control days and between large fire days and control days. Mahalanobis distances were calculated both for the whole dataset and separately for each season and region as:

$$Md = [(X_f - X_c)/\sigma]^2 \tag{1}$$

$X_f$ = mean fire-day index/code value, $X_c$ = mean control day index/code value; $\sigma$ = standard deviation of index/control value on combined fire and control days. All calculations were completed in R.

Our percentile analysis develops that described by *Andrews, Loftsgaarden & Bradshaw (2003)* by taking advantage of the paired nature of our large dataset (i.e., fire and control day observations with the same spatial and temporal distribution). *Andrews, Loftsgaarden & Bradshaw (2003)* compared fire days with all days in their dataset (i.e., fire days and non-fire days combined). We examined this approach but also examined two other related approaches:

1. We ranked the indices for control and fire days combined and examined the 90th, 50th and 25th percentiles of each index/code on fire days with those for control days.
2. We fitted a cubic spline curve (function "smooth.spline" in R) to model the control-day rank of each FWI System output as a function of the output's respective value. We then used the fitted model to predict what the control-day rank of each fire-day output value would be. By subtracting predicted rank from the actual rank we were able to examine the rank shift across the percentile position of all recorded fires. Larger differences in the percentiles indicated stronger discriminatory power.

Our paired data structure provides a more powerful test of the discriminatory power of the FWI system outputs whilst examining all percentile values provides a more complete assessment of output discriminatory ability across all the fire weather conditions captured, independent of the scale and distribution of the index.

Finally, we used Thiel-Sen regression analyses to model the relationship between the index/code percentile scores on fire-days and the rank of each percentile value across the fire days (*Eastaugh, Arpaci & Vacik, 2012*). The slope and intercept of the regression of percentile score on percentile rank were used as indicators of index/code performance where *Eastaugh, Arpaci & Vacik (2012)* claimed that a perfect discriminator would have a slope approaching zero and an intercept approaching 100. Smaller slopes and larger intercepts were therefore assumed to indicate better index performance. This method is, however, sensitive to both the overall number of days in the dataset and the number of fire days. This makes consistent interpretation of the results between different datasets difficult (see Supplemental Information 1). This was not a particular issue for us as we used a single dataset, but it did mean that comparisons of performance for all fire days and large fire days were not equivalent, we therefore only examined the former. In addition, with a very large number of fires, our data resulted in rather small slope values being estimated which made
it difficult to differentiate model performance using this metric. Thiel-Sen regressions were fitted in R 3.1.2 using the "mblm" function of the mblm package (*Komsta, 2013*).

Having selected the best performing FWI system outputs, we used logistic regression to model the probability of wildfire occurrence as a function of the chosen output and region, season, and location (urban/rural). For this analysis we split the data into test and training datasets where the test dataset was a 10% random sample of the full data and the training data the remaining 90%. Using the training data we initially fitted a full model which included all the main effects and all two-way interactions. The full model was then simplified by deleting non-significant terms ($P > 0.05$) and examining the change in Akaike's Information Criterion (AIC). Interaction terms were removed first and simplification stopped once deletions no-longer yielded further reductions in AIC >2. Models were fitted using the "glm" function in R 3.1.2 and specifying a binomial distribution and logit link function. Model performance was assessed using the test dataset by calculating the c-index, area under the receiver operating characteristic (ROC) curve (*Fawcett, 2006*). ROC curves illustrate the relationship between the false positive and false negative rate as the threshold probability value is changed. A c-index value of 0.5 indicates random predictions whilst a value of 1.0 indicates perfect performance. We used the same procedures to examine the ability of selected variables to discriminate between high magnitude fire days from normal (i.e., magnitude < 4) fire days. This latter analysis examines the extent to which the independent variables can model the probability of an observed ignition developing into a large fire.

## RESULTS

Wildfire occurrence varied substantially between regions and seasons with fires most common in the Highlands and during spring (Fig. 1). Patterns of fire magnitude also varied greatly between regions with data from Grampian being somewhat unusual as very few low magnitude (<3) events were reported (Fig. 1). Fires were more common in rural locations as were large magnitude events. There were slightly less wildfires during the legal managed burning season compared to outside it. The end of the legal burning period in spring was marked by a slightly increased density of wildfire occurrences but this period was also associated with somewhat higher than average FFMC values. There was no obvious increased density of occurrences at the start of the burning season in autumn (see Fig. S1 in Supplemental Information 2). Larger magnitude (>3) events occurred most frequently in spring and summer and in rural locations.

Initial screening of the FWI system outputs suggested the Initial Spread Index (ISI) and FFMC had the greatest discriminatory power (Fig. 2). Mahalanbois distances showed FFMC to be the best predictor overall, for large fires and for the Lothian and Highlands regions (Table 2). ISI performed best in the Dumfries and Grampian regions. FFMC performed best across all seasons except for the winter where the Duff Moisture Code (DMC) performed better. No output performed particularly well in summer, or in the Lothian region.

Similar results were obtained from the percentile analysis which revealed ISI and FFMC as the best predictors of fire occurrence in general whilst FFMC was marginally better at
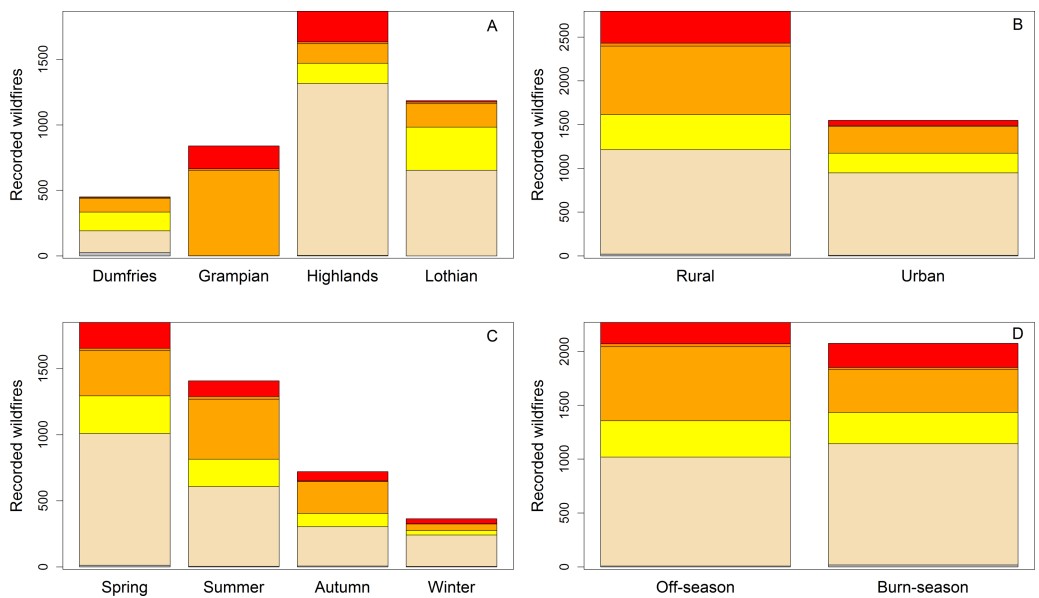

**Figure 1** **Spatial and temporal variation in the occurrence of Scottish wildfires according to, former Fire Brigade Region (A); urban and rural locations, where urban locations are within 10 km of one of Scotland's 100 larges towns and cities (B); season (C); and whether the fire occurred during the legal muirburn season (D).** Colours relate to fire magnitude, grey = 0, cream = 1, yellow = 2, orange = 3, dark orange = 4, red = 5.

predicting large fire events (Table 3). Differences in the results of *Andrews, Loftsgaarden & Bradshaw*'s (*2003*) percentile analysis and our modified paired version were minimal. The analysis of percentile rank shift (Fig. 3) confirmed the strong performance of the FFMC and ISI. Both indices showed strongest performance at intermediate levels of these outputs with FFMC's discrimination declining at higher levels. The related DMC and BUI indices showed similar patterns in discriminatory ability but considerably worse performance than the FFMC, ISI or FWI. During periods with the most extreme fire weather (>90th percentile) the DC appeared to provide marginally better discrimination.

Thiel-Sen analysis yielded somewhat different results to the previous scoping methods. The slopes of the regressions were all rather similar and close to zero due to the large number of days in the whole dataset (Table 4). Overall ISI was rated as the best discriminator of wildfire occurrence, but FFMC and FWI were more difficult separate with FWI having the better slope and FFMC the better intercept.

We selected FFMC and ISI as the two best performing outputs during scoping and these were used to model the probability of wildfire occurrence.We also tested the ability of FFMC, and ISI to discriminate between normal and high magnitude fire days. The final occurrence model using FFMC (AIC = 9389) had a c-index of 0.77 for the test dataset. The model included the main effects of FFMC, urban/rural location, season and region. There were significant interactions between FFMC and urban/rural location, FFMC and season and FFMC and region. The ISI model (AIC = 9429) had a c-index of 0.76 for the test dataset. The model included the main effects of ISI, urban/rural location, season and region. There were significant interactions

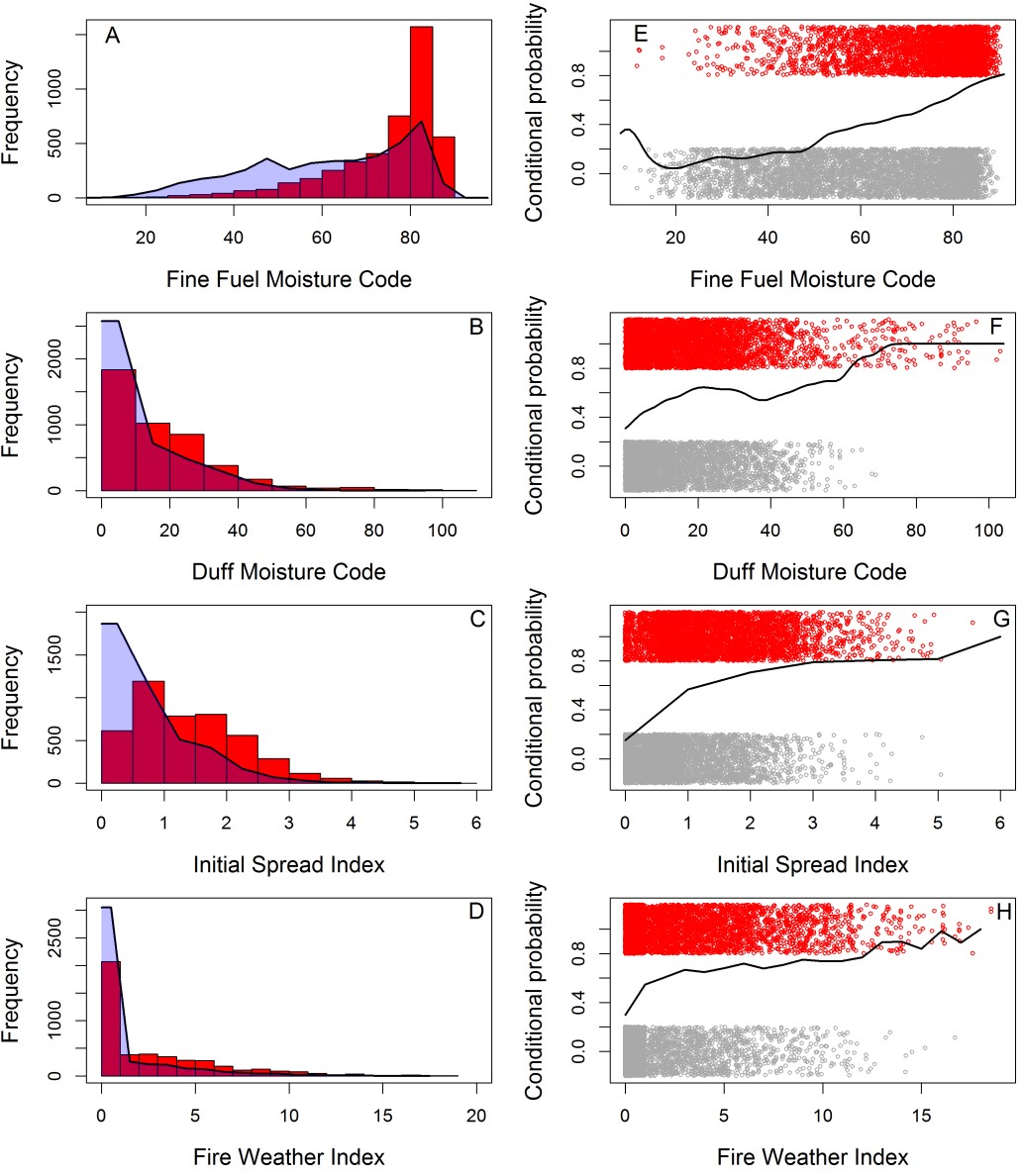

**Figure 2  Wildfire occurrence in four regions of Scotland in relation to key moisture codes and fire weather indices of the Canadian Fire Weather Index system.** The histograms (A–D) show fire occurrence in relation to (from top), FFMC, DMC, ISI and FWI. The distribution of moisture code/fire behavior index occurrences on control days is shown as a shaded polygon. The plots (E)–(H) show the conditional probability of a wildfire occurring across the range of FFMC, DMC, ISI and FWI in the data. The distribution of fire occurrences (red) and control days (grey) are also shown. Fire and control occurrences have been "jittered" on the y-axis.

between ISI and urban/rural location, ISI and season and ISI and region. Full model results for all analyses are reported in the Supplementary Tables. The modeled results (Figs. 4 and 5) suggested higher probabilities of wildfire at low FFMC or ISI values in autumn and winter and in urban areas. There was a relatively low probability of wildfire in Grampian at low FFMC values compared to the other regions. In all regions wildfires occurred at

**Table 2** **Mahalanobis distances describing the discriminatory power of moisture codes and fire weather indices for the wildfire dataset as a whole, for large fires only and for each region and season in the data.** The best performing code or index is shown in bold.

|  |  | FFMC | DMC | DC | BUI | ISI | FWI |
|---|---|---|---|---|---|---|---|
|  | Overall | **0.66** | 0.20 | 0.00 | 0.16 | 0.58 | 0.30 |
|  | Large fires | **1.16** | 0.32 | 0.01 | 0.23 | 1.06 | 0.52 |
| Region | Dumfries | 0.76 | 0.38 | 0.01 | 0.29 | **0.80** | 0.52 |
|  | Grampian | 0.67 | 0.52 | 0.01 | 0.44 | **0.72** | 0.65 |
|  | Highlands | **0.84** | 0.15 | 0.00 | 0.10 | 0.68 | 0.23 |
|  | Lothian | **0.39** | 0.12 | 0.00 | 0.10 | 0.34 | 0.19 |
| Season | Spring | **1.09** | 0.65 | 0.03 | 0.61 | 0.98 | 0.63 |
|  | Summer | **0.35** | 0.21 | 0.01 | 0.21 | 0.33 | 0.32 |
|  | Autumn | **0.56** | 0.45 | 0.10 | 0.46 | 0.54 | 0.52 |
|  | Winter | 0.61 | **0.82** | 0.01 | 0.62 | 0.78 | 0.68 |

**Table 3** **Above: Fire Weather Index system code and index values for the 25th, 50th and 90th percentiles for control days/combined fire and control days ('all' days); Below: difference between the fire day percentile and the control/all day percentile and for (A) all fire days and (B) large fire days.** Values indicating the best performing code/index for each percentile are shown in bold.

| Percentile | FFMC | DMC | DC | BUI | ISI | FWI |
|---|---|---|---|---|---|---|
| **All/Control Day** |  |  |  |  |  |  |
| 25th | 48.0/57.9 | 1.7/2.8 | 35.4/45.7 | 2.6/4.5 | 0.1/0.4 | 0.1/0.2 |
| 50th | 64.4/74.4 | 6.0/9.7 | 200.1/199.8 | 10.6/15.8 | 0.6/0.9 | 0.4/0.7 |
| 90th | 82.8/84.6 | 30.3/33.9 | 533.9/545.4 | 51.2/57.1 | 1.8/2.3 | 4.5/6.3 |
| **(A) Fire Day difference from control/all day percentile** |  |  |  |  |  |  |
| 25th | **21/15** | 16/9 | 7/3 | 16/8 | **21/15** | 20/14 |
| 50th | **33/18** | 21/11 | 0/0 | 19/10 | **33/17** | 30/15 |
| 90th | **22/6** | 7/3 | 5/3 | 6/3 | **22/6** | 16/5 |
| **(B) Large Fire Day difference from control/all day percentile** |  |  |  |  |  |  |
| 25th | **24/20** | 20/13 | 5/1 | 18/10 | **24/20** | 23/19 |
| 50th | **43**/28 | 21/12 | −4/−4 | 18/10 | 42/**29** | 37/22 |
| 90th | 31/11 | 11/7 | 5/3 | 11/6 | **34/14** | 21/10 |

noticeably higher values of ISI in summer. The probability of wildfire increased rapidly at FFMC values >50 and was generally very high (>80%) at ISI values >2.

FFMC and ISI had approximately equivalent ability to distinguish between high magnitude fire events and "normal" fire days with c-index values of 0.76 and 0.77 respectively. Their performance in detecting large fires was also roughly equal during scoping. Overall we preferred the FFMC model for the sake of consistency and because it performed slightly better during the final modeling process. The final model included the main effects of FFMC, region and season and an interaction between season and FFMC (see Supplementary Tables; Fig. 6).

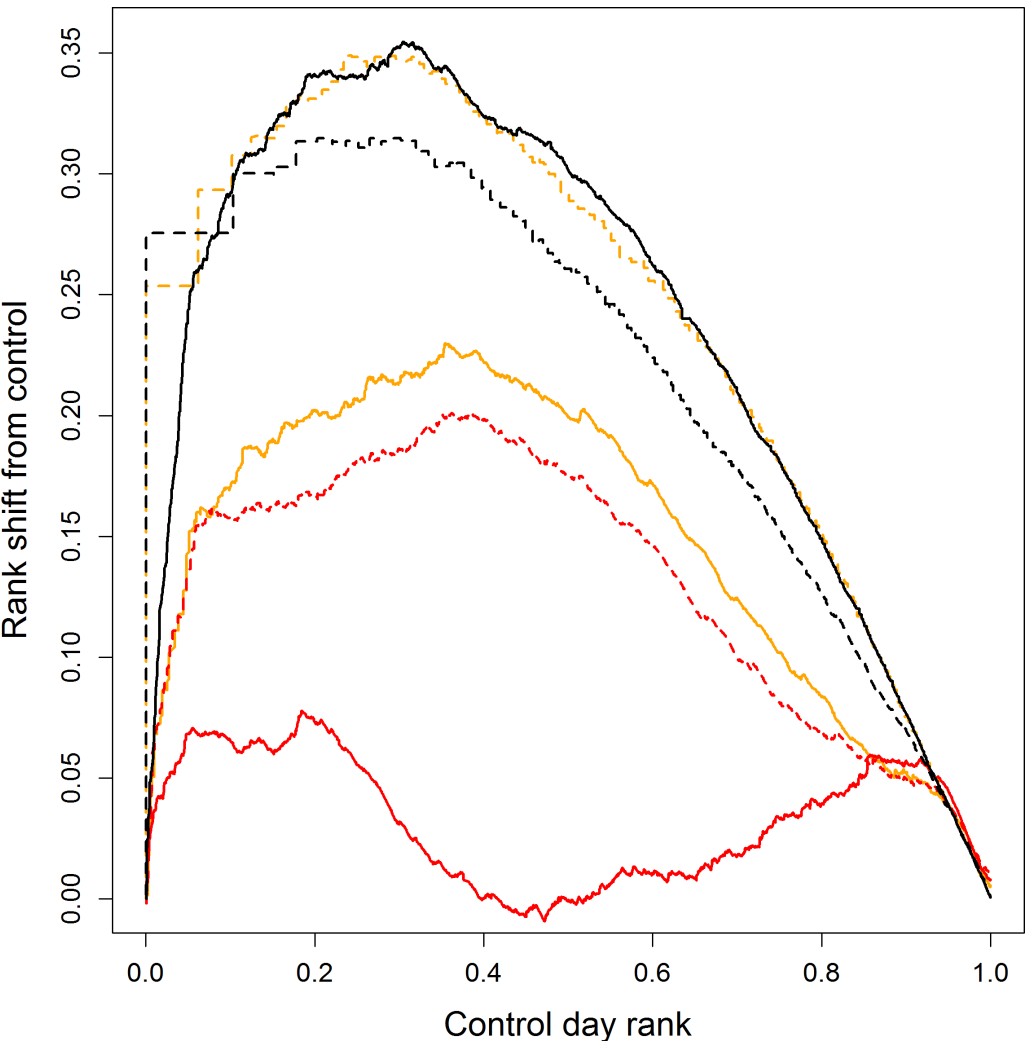

**Figure 3** **Rank shift between fire and control days for FWI system outputs' percentiles for wildfires in Scotland.** The analysis compares the difference in rank between that recorded on a fire day and the rank of an equivalent value on control days. Higher values represent greater discriminatory power. Each output is shown in a different colour/style: FFMC, solid black; DMC, solid orange; DC, solid red; ISI, dashed orange; BUI, dashed red; and FWI, dashed black.

## DISCUSSION

Patterns of wildfire activity vary significantly across Scotland with the most wildfires recorded during the study period occurring in the Highlands and Islands region. However, the four different study regions differ substantially in both area and population meaning that whilst on a per area basis Highlands and Islands sees the least wildfires, per person it receives by far the most (Table 5). The opposite is true for the more densely populated L&B region that includes the city of Edinburgh and its outlying towns and villages. The trends are likely to be driven in part by differences in climate, land-cover, land-management and burning practice. Much of sparsely-populated western Scotland is managed extensively for sheep and deer grazing and traditional burning practices are associated with relatively large

**Table 4  Sumary of Thiel-Sen analyses on the discriminatory power of codes and indices of the Canadian Fire Weather Index system.** The columns Rank Intercept and Rank Slope show the rank order of code/index performance according to each of the metrics.

| Fire weather variable | Intercept | Slope | Rank Intercept | Rank Slope |
|---|---|---|---|---|
| FFMC | 23.27 | 0.019 | 2 | 3 |
| DMC | 15.65 | 0.020 | 4 | 5 |
| DC | −0.12 | 0.023 | 6 | 6 |
| BUI | 13.33 | 0.020 | 5 | 4 |
| ISI | 23.37 | 0.019 | **1** | 2 |
| FWI | 22.04 | 0.019 | 3 | **1** |

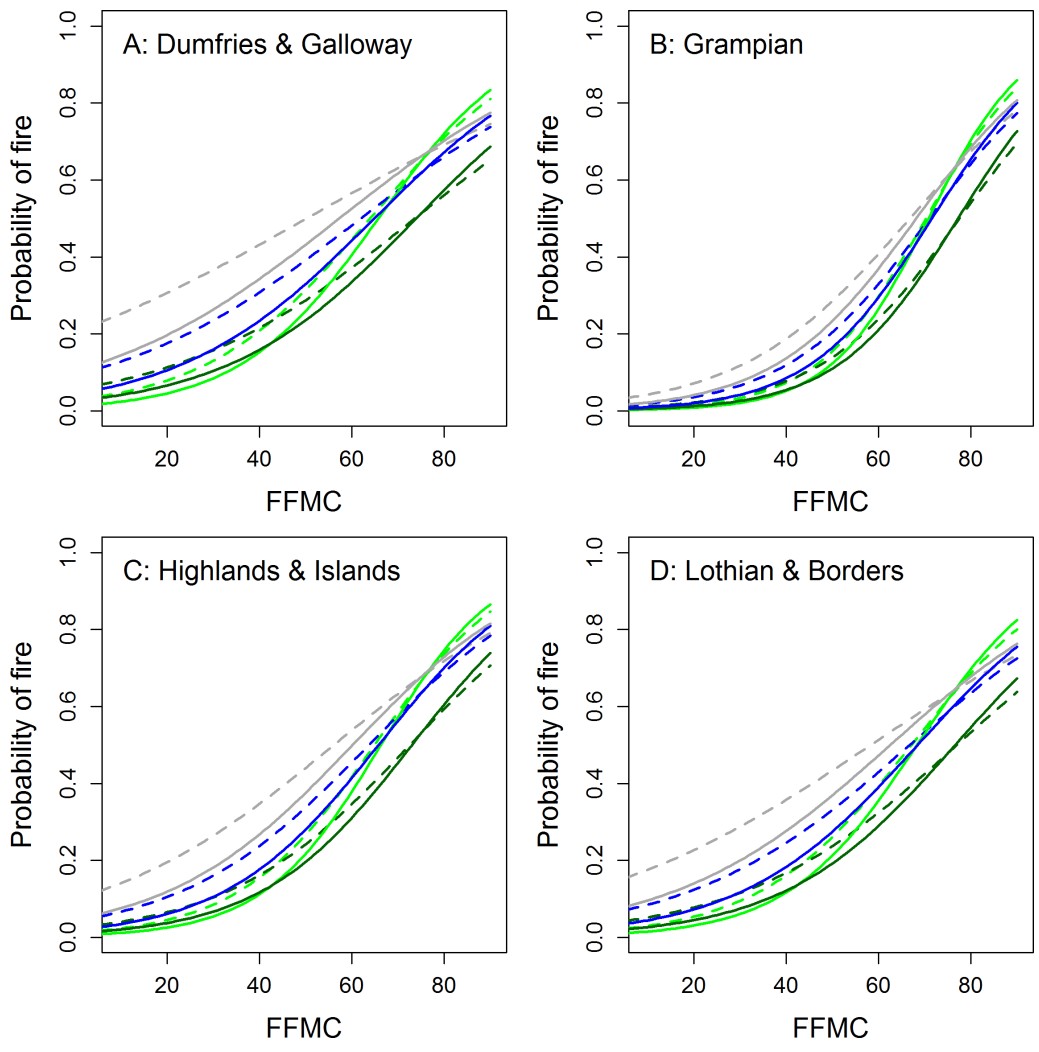

**Figure 4  Modelled probability of wildfires in Scotland as a function of FFMC and season for rural (solid lines) and urban (<10 km from one of Scotland's 100 largest towns or cities; dotted lines) locations across four regions of Scotland.** Green, spring; dark green, summer; blue, autumn; grey, winter.

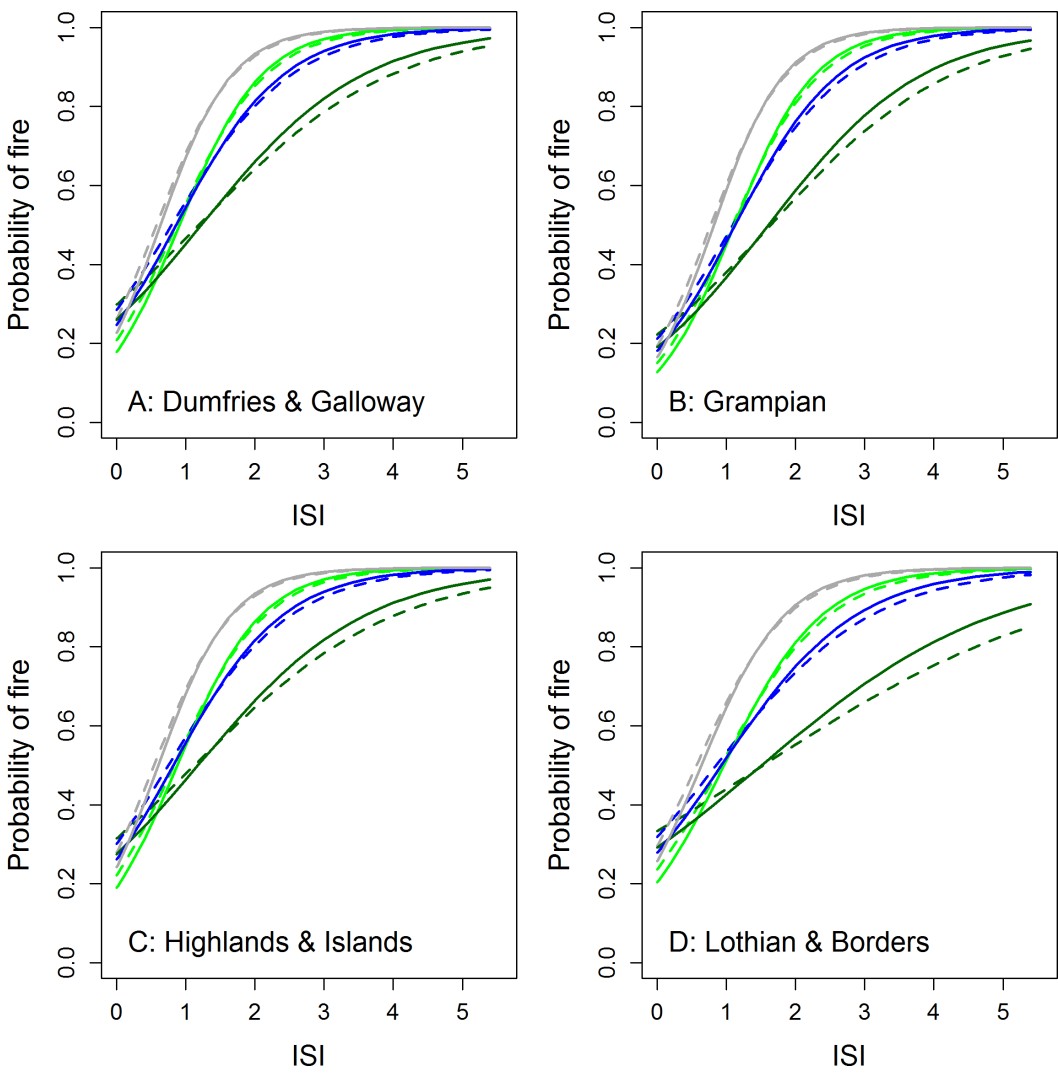

**Figure 5** **Modelled probability of wildfires in Scotland as a function of ISI and season for rural (solid lines) and urban (<10 km from one of Scotland's 100 largest towns or cities; dotted lines) locations across four regions of Scotland.** Green, spring; dark green, summer; blue, autumn; grey, winter.

and uncontrolled burns to improve forage quality (*Hamilton, 2000*). Grampian is a key area for traditional moorland management for grouse shooting (*Douglas et al., 2015*) and it is noticeable that here, where intensive prescribed burning occurs each spring, wildfires are more common. There are, however, also strong gradients in fuel type and climate from west to eastern Scotland with the shrub-dominated moorland fuels and drier climate of eastern Scotland making fire hazard somewhat higher. On a per area basis wildfires were most common in Lothians and Borders. Though fire is used in the southern upland ecosystems of the Scottish Borders, the dense population of the Lothians means a substantial number of smaller urban-interface fires in, for example, stands of gorse may account for a large number of these fires. Though, overall, fewer fires were recorded in urban than rural locations (Fig. 1) the total urban area is a small proportion of the land area of Scotland.

**Table 5  Regional variation in wildfire activity during the study period (1st January 2003–15th March 2007) in relation to land-surface area and population.** The time-span of the L&B data was 7 months shorter than the other regions so these figures are conservative in comparison to the other regions.

| Region | Wildfires per km² | Wildfires per person |
|---|---|---|
| Dumfries & Galloway | 0.07 | 0.003 |
| Grampian | 0.10 | 0.002 |
| Highlands & Islands | 0.05 | 0.007 |
| Lothian & Borders | 0.18 | 0.001 |

The tentative conclusions we can draw above are somewhat dependent on the quality of the data. Though broad locations for fires are recorded these often have low precision and little supporting information such as the cause of the fire, area and predominant fuel type. Efforts to improve the quality of data recording are underway (e.g., *Gazzard, 2009*) and, along with the unification of Fire and Rescue Services, this may solve some of the issues. In this context it was also noticeable that the wildfires recorded in Grampian, a core area for traditional managed burning for red grouse, were nearly all of medium-high magnitude. There are a number of possible explanations for this including that Grampian may tend to deploy more resources to fires than other regions or that smaller ignitions may not have been recorded (perhaps because land-managers tend to extinguish small wildfire themselves). If the latter is the case then the incidence of fire occurrences has been underestimated there. Better collaboration between the FRS and researchers is an urgent need across the UK and developing a robust fire danger rating system will require better quality data than we had to work with here.

Despite the broad spatial scale at which fire-weather forecasts were made, the FFMC and ISI performed tolerably well as predictors of wildfire occurrence. Our results for Scotland are similar to those found in other studies in the UK. *Albertson et al. (2009)*, working in the Peak District National Park, found that wildfire occurrence was positively related to maximum temperature, reduced rainfall and days when visitor numbers were likely to be higher. Furthermore, De Jong et al. (2015) used a percentile-based approach to assess the performance of the FWI System across the UK and found large regional differences in the numeric values of code/index 99th percentiles. Their results also suggested that the FFMC and ISI held promise for forecasting wildfire activity. De Jong et al.'s and our results are interesting as previous research on fire behaviour in heather-dominated upland ecosystems in Scotland found little relationship between the outputs of the FWI system and fire rate of spread and intensity (*Davies et al., 2006*). This may be because the rapid moisture response times of fine, elevated, dead shrub and grass fuels mean standard forms of the FWI system codes are not appropriate for predicting their FMC (*Legg et al., 2007*; *Anderson & Anderson, 2010*). Unlike in other European shrubland ecosystems (e.g., *Castro, Tudela & Sebastià, 2003*; *Pellizzaro et al., 2007*) live fuel moisture is not adequately reflected by the DMC and DC as physiological drought conditions are rare in Scotland's wet organic soils and moisture dynamics are driven by seasonal variation in plant physiology (*Davies et al., 2010*). We have little published information on the relationship between *Molinia* litter moisture content and fire weather. As with Scottish shrub fuels, summer drought

and "curing" of vegetation are not particularly important—rather seasonal patterns of flammability are associated with the autumn-die back of grasses in response to changes in day length (*Salim et al., 1988*). Notwithstanding this, particularly high values of the DMC and DC may relate to the potential for smouldering of deeper duff and organic soil layers and thus higher severity fires (*Davies et al., 2013*; *Davies et al., 2016b*).

This begs the question of why is the FWI system able to predict wildfire occurrence but not fire behaviour? One important reason is likely to be that, for safety reasons, experimental fires have not been completed over a particularly wide range of fire weather conditions and thus exclude those associated with wildfires. However, we posit that the particular structural characteristics of moorland fuel types may also be important and that key variables governing fuelbed ignitability and fire behaviour differ. This is similar to the conclusions of *Alexander (2008)* who suggested that fuel moisture acted as an "on/off switch" in shrub fuel types. An important difference in Scottish moorlands compared to other fire-prone regions is that vegetation often grows on saturated organic soils or peat that, for much of the year, retain moisture contents high enough to prevent their ignition. The heather canopy and *Molinia* litter are readily flammable despite the former containing large proportions of live vegetation (*Davies et al., 2008*; *Davies & Legg, 2011*; *Santana & Marrs, 2014*). Moorland fuels have surface layers of shrubs and grass underlain by pleurocapous mosses, *Sphagnum* spp. and plant litter. During typical managed burning activities, and many wildfires, these fuels are often too wet to burn (*Davies et al., 2016b*). Critically therefore, many fires only burn through the heather canopy or the *Molinia* litter and the high moisture content of the moss and litter actually plays an important role in controlling the efficacy of the traditionally-used "firebeaters" to control and extinguish burns. In the case of heather-dominated fuels, and other shrublands (e.g., *Anderson & Anderson, 2009*), their behaviour has been viewed as akin to mini independent crown fires.

The flammability of the heather canopy and *Molinia* litter, and their ability to burn at high intensities even under very low FFMC/ISI values (*Davies et al., 2006*), means that wildfire activity may often only be limited by the lack of a suitable ignition source. Ignition potential of the heather canopy has been shown to be a function of the fuel moisture content of dead vegetation in the lower canopy (*Davies & Legg, 2011*), whilst the most important controls on managed fire behaviour appear to be fuel structure, wind speed and live fuel moisture content (*Davies et al., 2009a*). Previous research has demonstrated that FMC also plays an important role in determining fire behaviour in *Molinia* litter fuels (*Hamilton, 2000*). The lack of a relationship between observed fire behaviour and FWI system fire behaviour indices has been ascribed to the poor predictive power of the underlying moisture codes for shrub fuels (*Anderson & Anderson, 2009*; *Davies et al., 2006*). However, moss and litter layers contain as much, if not more fuel, than surface layers (*Davies et al., 2008*) and if flammable will contribute significantly to increases in fire rate of spread (*Davies & Legg, 2011*), intensity and control difficulty. They and *Molinia* litter fuels have been shown to be readily ignitable from small, smouldering ignition sources (*Davies et al., 2009b*) when sufficiently dry. The probability of ignition of the heather canopy by such mechanisms may be low, not only as such ignition sources are likely to fall through it into the moss and litter, but also because rates of heat loss will be high, and contact between

the ignition source and fuel low, making successful ignition unlikely. Previous research (*Legg et al., 2007*; *Grau et al., 2015*) has suggested that the FFMC is correlated with the moisture content of moss and litter fuels possibly as they are similar in composition and structure to the forest litter layers for which the FFMC was developed. Microclimatic conditions below heather or grass canopies may also approximate those found on the forest floors for which the FFMC was designed.

Given the above, we can summarize a conceptual model of factors allowing the ignition of wildfires in moorland fuels, and their prediction by the FWI System as follows:

1. Accidental fires originate from weak ignition sources that are unlikely to ignite shrub canopies. They are thus primarily a function of the moisture content of the moss and litter layer. The moisture content of this layer is correlated with the FFMC in most seasons allowing it to predict ignition potential. In winter, during periods of reduced day-length, higher precipitation and cold temperatures, the DMC performs better as its slower response more adequately represents the longer rainless periods required for moss/litter flammability.

2. Moisture conditions suitable for ignition and spread of fires ignited by strong ignitions (e.g., from a drip-torch or deliberate arson attempts) are not well-represented by the FFMC due to canopy fuels' rapid moisture response times and the marginal fire weather conditions under which such fuels can burn. However, flammability of the moss/litter layer plays a role in fire controllability and is associated step-changes in fire behaviour. The FFMC thus detects periods when managed fires are more likely to escape control.

Our scoping studies and the logistic regression analyses also revealed that there was substantial spatial variation in fire risk in relation to fire weather between regions and seasons (Table 2 and Figs. 4–6). Wildfires were more likely in urban areas during low-moderate risk conditions but are less likely when wildfire risk is very high. Fires are also more likely during marginal conditions in winter and autumn. This would again seem to suggest that wildfires are strongly ignition limited. From Fig. 1, wildfires appeared to be no more likely during the legal burning season than at other times but given this period includes the winter months, which are frequently very wet or have snow cover, the number of actual days conducive to fire may be somewhat limited and fire activity higher on a per available day basis. Figure S2 also suggests a clustering of wildfire activity around the end of the burning season in spring. This may be partly caused by apparently drier conditions during this period but human factors and management decisions may also be at play. Exploring this question will require a better knowledge of the relationship between fire weather conditions and ignition potential (e.g., *Tanskanen et al., 2005*) and collaborative work with fire managers that encourages them to report conditions under which they face control difficulties.

The FFMC, ISI and, to a lesser extent FWI, were also able to discriminate between high magnitude wildfires and smaller scale events fairly well. The modelled probability of an occurring fire being "large" must be treated with caution as the model is dominated by the effect of Region (see Supplementary Tables) which may be at least partly related to differences in the completeness of fire recording between regions (Fig. 1). The probability of a fire being large was generally low, though higher in the more remote, northerly Grampian

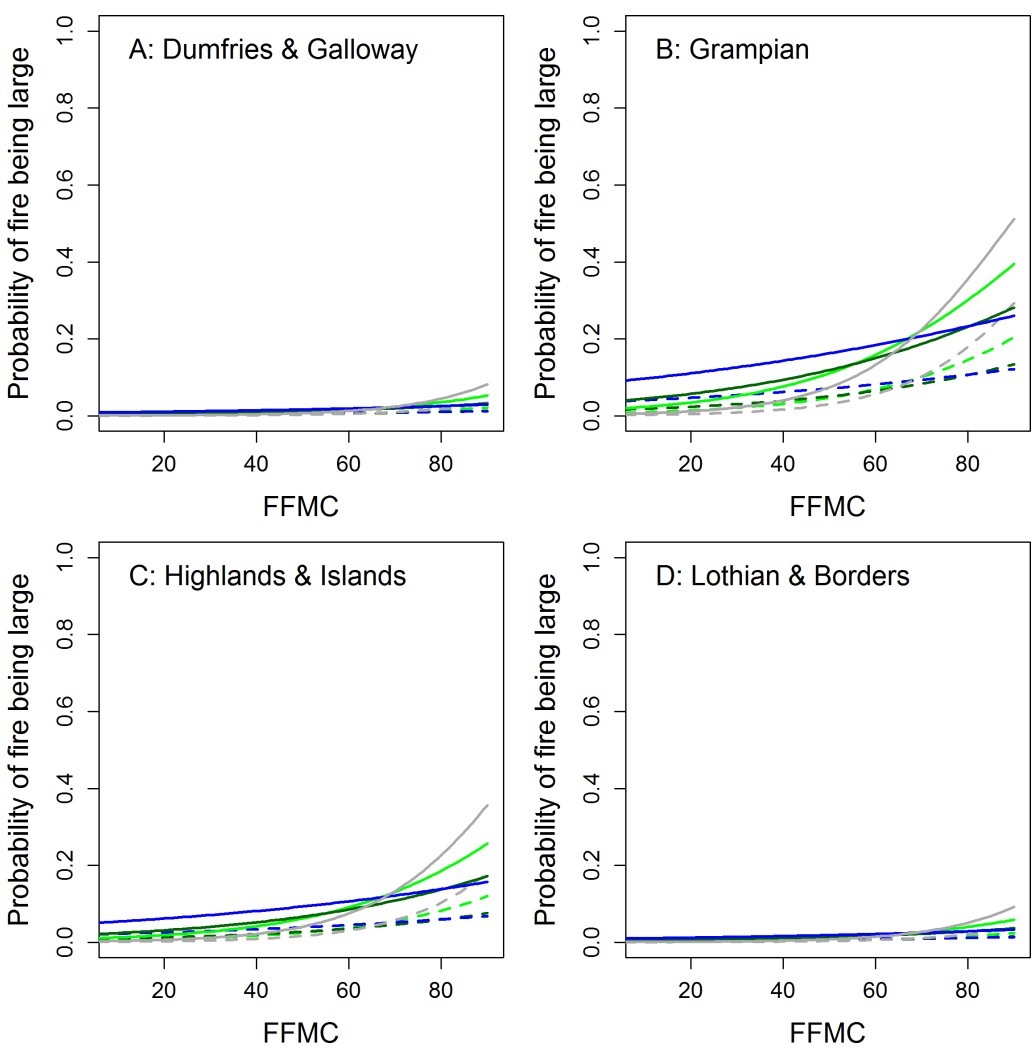

**Figure 6  Modelled probability of observed wildfires in Scotland being "large" as a function of FWI and season for rural (solid lines) and urban (<10 km from one of Scotland's 100 largest towns or cities; dotted lines) locations across four regions of Scotland.** Green, spring; dark green, summer; blue, autumn; grey, winter.

and Highlands regions (Fig. 6). The probability of a wildfire becoming a large may thus be a function of accessibility. Remote fires of any size are less likely to be reported early and will be large by the time the Fire and Rescue Services have arrived on the scene. Very few wildfires occur in winter, a time when FFMC values are generally low, but for higher values of FFMC the probability of a large wildfire was greatest during this season. The few large wildfires recorded during this period are clustered towards the end of February when FFMC is beginning to increase (see Fig. S2) and wildfire preparedness may be low.

Scotland remains some way off developing an operational fire danger rating system and the difficulties encountered by countries with similar fuels and climates (e.g., *Alexander, 2008*) means considerable research is needed. The need for a fire danger rating system is growing as climate change is likely to increase the risk of both wildfires in general (e.g.,

*Scholze et al., 2006*; *Sutherland et al., 2008*) and severe, damaging fires associated with the smouldering of ground fuel layers and peat (*Maltby, Legg & Proctor, 1990*; *Davies et al., 2013*). Scotland currently lacks its own system and is not covered by the Met Office Fire Severity Index which provides a forecast of severe fire weather conditions. Coarse-scale forecasts of the FWI system outputs are, however, available from the European Forest Fire Information System (http://forest.jrc.ec.europa.eu/effis). Choosing an appropriate index to forecast fire risk is complicated as different indices performed better in different regions, at different times of year and for different purposes. Further research using longer spans of data are urgently needed and should also seek to link fire weather controls on wildfire risk to an analysis of the spatial distribution of ignitions in relation to vegetation types and anthropogenic activity (e.g., *Oliveira et al., 2012*; *McMorrow & Lindley, 2006*). The FRS or other relevant agencies must invest in research and provide a regular stream of fire data to researchers if they want to develop a robust system. We also urge the Met Office to provide UK wide forecasts of the actual FWI system outputs so managers can make informed decisions.

## CONCLUSIONS

The wildfire probabilities we've presented must be treated with caution as they are only the probability of a reportable wildfire in the presence of an anthropogenic ignition (*Finney, 2005*). In the absence of any other information they could, in conjunction with EFFIS forecasts, be used by managers and the FRS to provide some advance warning of the potential for increased wildfire activity. On the basis of our results we suggest the following rules of thumb:

- Managed burning be avoided when FFMC > 75 as predicted probabilities of wildfire are generally >60% during such periods (Fig. 4).
- Fire-fighting resources be placed on stand-by when ISI approaches 2 as in most regions and seasons the probability of wildfire will be > 80% (Fig. 5).
- Managed burning be avoided and fire-fighting resources on stand-by when DMC > 60 as, though these conditions are rare, nearly all days in our dataset associated with such conditions were fire rather than control days (Fig. 2). Burns during such conditions are likely to have high fire severities.
- In northern Scotland the Fire and Rescue Services should be prepared for higher magnitude wildfire events when FFMC > 75 as the probability of a wildfire being "large" increased steeply above this value (Fig. 6).

## ACKNOWLEDGEMENTS

We are indebted to the Scottish Fire and Rescue Service for the provision of their wildfire data. Karl Kitchen and Penny Marno of the Met Office facilitated access to the NWP weather data used to generate the FWI system outputs. Michael Bruce, Jeff Ord and Stuart Anderson offered many helpful insights. Wendy Anderson provided a detailed and constructive peer-review of the initial research report from which this paper developed.

Paulo Fernandes and two anonymous reviewers provided many insightful comments which improved the paper.

### Funding
Funding for this research was provided by the Scottish Government through the Scottish Wildfire Forum and by Scottish Natural Heritage. The funders had no role in study design, data collection and analysis, decision to publish, or preparation of the manuscript.

### Grant Disclosures
The following grant information was disclosed by the authors:
Scottish Government.

### Competing Interests
G. Matt Davies is an Academic Editor for PeerJ.

### Author Contributions
- G. Matt Davies conceived and designed the experiments, performed the experiments, analyzed the data, wrote the paper, prepared figures and/or tables, reviewed drafts of the paper.
- Colin J. Legg conceived and designed the experiments, performed the experiments, reviewed drafts of the paper.

### Data Availability
Data on wildfire occurrences are the property of the Scottish Fire and Rescue system. Their permission is required for further dissemination. This can be organised by contacting the authors.

### Supplemental Information
Supplemental information for this article can be found online at http://dx.doi.org/10.7717/peerj.2649#supplemental-information.

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
