# Peer review of "Regional variation in fire weather controls the reported occurrence of Scottish wildfires"

_PeerJ, doi:10.7717/peerj.2649_

## Round 0.1 · original submission · Major Revisions

All three reviewers provided relevant input and suggestions to improve your article, please, address each of these points carefully in your rebuttal.

·

Basic reporting

The manuscript conforms to the basic reporting guidelines.

Experimental design

All standards are met.

Validity of the findings

All standards are met.

Additional comments

The manuscript addresses fire danger rating, a cornerstone of modern fire management that remains insufficiently researched and developed in Europe. The work reads very well and is devoid of apparent flaws. In particular I did appreciate the combination of different statistical methods to analyse data. I have only a few requests for corrections and clarifications and an interpretation issue.

L29. Isn’t “did a tolerable job” too colloquial?

L55. Add “fires” after “wildland”.

L83-94. “considerable” used 4 times.

L106. Not clear what was modeled.

L125. Does “event magnitude” mean “fire size”? Would be more clear to say so or explain a bit.

L210. I understand that in contrast to other studies your “fire day” approach is for a specific location rather than a region but why didn’t you use FWI data for the entire set of days? Can the weather of a single day be considered typical?

L261-267. I see this content better placed in the Discussion.

L310. The DC appeared to allow marginally better discrimination?

L314. More difficult to separate.

L375. A plausible explanation is that the functional forms of fire behaviour equations in the Canadian FFBPS (which were derived essentially for forests and dead fuels) are not suited for shrub fuels.

L390-392. This suggestion does not agree with the results, otherwise the FWI and/or DMC/BUI would have scored better than the FFMC and ISI. It is also not consistent with current knowledge of fire behaviour drivers in shrubland and grassland, which are essentially wind and dead fuel moisture content, e.g. for the former see Anderson et al. (2015), IJWF. See also p. 10 on Alexander (2008) for a sound explanation, even though I recognize that Calluna shrublands fuel structure differs from other shrubland types.

L396-398. Too colloquial? Anyway I suggest deletion as it does not add relevant content. Whether or not an ignition source exists fuel conditions will determine if fire can ignite and spread.

L407. Again, if that is so why doesn’t the DMC or FWI perform better? An alternative explanation can be offered if indeed litter/moss fuels are relevant for fire spread: the FFMC was developed for forests and so it will not account for the faster changes in dead fuel moisture content possible in open environments (see Cruz et al. 2016, IJWF for the FFMC-fuel moisture content relationship in grassland and modified FFMC for shrubland of Anderson & Anderson 2009). The fact that the litter/moss layer is shaded, at least partially, will make it more similar to forest litter in the response to atmospheric conditions hence explaining the FFMC relevance.

L428. Because of lack of fire detection towers?

L430-431. Possibly because fire suppression preparedness is lower in winter?

L438. It would be interesting to discuss how the study results relate with those of de Jong et al. (2016).

L458. Maybe I missed it but I don’t recall seeing the DMC threshold mentioned in the results. By analogy with other vegetation types DMC=60 seems too high to carry out burning activities, because it will likely correspond to excessive (if not total) consumption of the organic layer.

Reviewer 2 ·

Basic reporting

The paper addresses wildfire management issues in Scotland. As it is, the innovation steams from the geographic context. Nevertheless, it is little more than a curiosity, with little overall scientific or societal impact, although it could be improved if a relation with the wildfire management strategies and tools in fire prone areas, was performed.

Experimental design

The experiemntal design has strong limitations. as recognized by the authors, which may hamper the quality of results and the validity of the findings. Furthermore, the Materials and Methods are extense, too extense, since part of the text could be placed at the "Results" chapter, but fails to define some of the indexes used in the paper.

Validity of the findings

The authors state that the database has problems that may influence the results quality.

Additional comments

The innovation of this paper lies on the geographic context, linked to the peculiarity of wildfire management in a temperate maritime climate. The way it is constructed, making little reference to the mainstream wildfire management theory at wildfire prone areas (e.g. Mediterranean USA, etc ), makes this paper little more than a curiosity that will only be referred to by the author, with little impact overall, and dragging down the Journal Impact factor.
And yet there are subjects that would be relevant to discuss. For instance, prescribed fire is used in Scotland, as in all fire prone regions, a management technic, apparently not very well considered. The UK, apparently has a fire suppression at all costs strategy, that has been abandoned at the most important fire prone areas worldwide. The discussion on why the UK still has a suppression at all cost strategy would be interesting and important.
The "Materials and Methods" section is too large, part of the text could be placed at the "Results" section, which is too small for such a paper. It fails, nevertheless to present the definition of the indexes used, which is altogether more surprizing since they play an important role in the analysis.
The paper would benefit from a localization map.
The authors question the quality of the database, which has implications for the quality of the analysis.
This together with the fact that the way the paper is written will cause little impact and is little more than a curiosity, since it is disentangled from current worldwide practice and theory, makes me think that the authors would gain if they would compare their experience and explain why some of the management options are opposite to what has been done for at least the last 2 decades in fire prone areas, where fire was reintroduced as a management tool, and strategies were developed to take advantage of this tool that has been used traditionally since primeval times, apparently also in Scotland, which is quite a surprize.

Reviewer 3 ·

Basic reporting

The manuscript is very clearly written, describing the background and rationale very clearly. Data is presented in sufficient depth. Figures are Tables are clear and necessary. The manuscript is thus sufficiently self-contained. The most relevant literature has been cited.

Experimental design

The submission is primary research within scope of Peer J and the research gap very clearly presented. The experimental design is clearly described and fundamentally sound. There are substantial uncertainties, assumptions and gaps in the approach, but the authors make it very clear that this work is of very exploratory nature. Given the importance of the research gap this exploratory approach provides a valuable contribution.

Validity of the findings

Much of the value of this study lies in highlighting a major problem, the associated gaps in data, the lack of a fire risk prediction system in Scotland (and essentially a useful one for the whole of the UK), and the presentation of some intriguing insights about actual fire occurrence and its relationship meteorological and environmental variables.
Much of the findings discussed are exploratory and tentative, however, the authors conclude with useful and potentially very impactful management threshold recommendations based on outputs from the Canadian Fire Weather Index System (FWI). The FWI is based on large amount of empirical observations in Canada and has proven to performing quite well in some other regions of the world. These suggested (rules of thumb) are likely to be useful and probably better than not having any, but they are out of line with the otherwise careful presentation and do require a more solid justification than is presently given.

As to the literature, there are some further studies, which have attempted the challenging task of linking fire weather to fire activity, which may provide some further useful insights.

C. C. Simpson, H. G. Pearce, A. P. Sturman, P. Zawar-Reza (2014)
Verification of WRF modelled fire weather in the 2009–10 New Zealand fire season. International Journal of Wildland Fire, Vol. 23 No. 1 Pages 34 – 45.

C.C. DaCamara, T. J. Calado, S.L. Ermida, I.F. Trigo, M. Amraoui, K.F. Turkman (2014) Calibration of the Fire Weather Index over Mediterranean Europe based on fire activity retrieved from MSG satellite imagery
International Journal of Wildland Fire, Vol. 23 No. 7 Pages 945 – 958.

Additional comments

See some minor additional comments in the annotated PDF

Annotated reviews are not available for download in order to protect the identity of reviewers who chose to remain anonymous.

---

## Round 0.2 · accepted · Accept

The substantial and adequate responses addressing the comments and suggestions of the various reviewers have resulted in a well structured, interesting, clear and publishable manuscript. Well done!.